# Breeding D1-Type Hybrid *Japonica* Rice in Diverse Upland Rainfed Environments

**DOI:** 10.3390/ijms26073246

**Published:** 2025-03-31

**Authors:** Chunli Wang, Juan Li, Qian Zhu, Junjie Li, Cui Zhang, Ruke Hong, Dajun Huang, Zhonglin Zhang, Jin Xu, Dandan Li, Jiancheng Wen, Chengyun Li, Youyong Zhu, Dongsun Lee, Lijuan Chen

**Affiliations:** 1Rice Research Institute, Yunnan Agricultural University, Kunming 650201, China; wchunli1989@163.com (C.W.); lijuan1661@126.com (J.L.); gabriel731@ynau.edu.cn (Q.Z.); ad9979li@163.com (J.L.); 15096694734@139.com (C.Z.); hlshrk@163.com (R.H.); hdj668@163.com (D.H.); zzl681128@163.com (Z.Z.); genexu@126.com (J.X.); lidanzaizhe@163.com (D.L.); jcwen1117@163.com (J.W.); 2State Key Laboratory for Conservation and Utilization of Bio-Resources in Yunnan, Yunnan Agricultural University, Kunming 650201, China; lichengyun@ynau.edu.cn (C.L.); yyzhu@ynau.edu.cn (Y.Z.); 3The Key Laboratory for Crop Production and Smart Agriculture of Yunnan Province, Yunnan Agricultural University, Kunming 650201, China

**Keywords:** hybrid *japonica* rice, unique adaptability, rainfed upland planting, diverse ecological zones

## Abstract

‘Dianheyou615’ (DHY615) is an elite Dian (D1)-type hybrid *japonica* rice variety, renowned for its high yield, exceptional grain quality, and unique adaptability to both irrigated and rainfed conditions in the Yungui Plateau of southwestern China. However, the genetic mechanisms underlying the agronomic performance of the D1-type hybrid *japonica* rice remain unclear. In this study, a comprehensive analysis of ‘DHY615’’s agronomic performance, genetic genealogy, and molecular genetic foundation was conducted to dissect its desirable traits for upland rainfed cultivation across diverse ecological environments. The main findings indicate that ‘DHY615’ possesses 6432 heterozygous SNPs, with 57.48% and 14.43% located in the promoter and coding regions, respectively, potentially affecting key phenotypic traits. High-impact SNPs variants and numerous well-known functional genes were identified, such as *OsAAP6*, *GS3*, *Sd1*, *Rf1*, *BADH2*, *BPh14*, *Rymv1*, *OsFRO1*, *NRT1.1B*, *SKC1*, *OsNCED2*, and *qUVR-10*, which are likely linked to traits including plant architecture, grain yield, grain quality, and resistance to various biotic and abiotic stresses (e.g., disease, cold, drought, salt, high iron, and high UV radiation). Notably, ‘Nan615’ harbors a greater number of functional allele variants compared to ‘H479A’, which potentially explaining its superior grain yield and remarkable adaptability. This study offers novel and valuable insights into the molecular genetic foundation of the plateau D1-type hybrid *japonica* rice, underscoring its potential for sustainable rice production across diverse ecological zones, especially with its unparalleled high-altitude adaptability to rainfed upland planting.

## 1. Introduction

Rice (*Oryza sativa* L.) is a typical monoecious and self-pollinated crop. The successful development of the three-line hybrid rice system (cytoplasmic male sterility or CMS (A-), maintainer (B-), and restorer (R-) lines) was the core genetic tool used to harness heterosis. Currently, the Wild–Abortive (WA) type and Hong Lian (HL) type are the primary CMS sources used in the production of the three-line hybrid *xian* (*O. sativa* ssp. *indica*), while the Boro–Taichung (BT) type and Dian I (D1) type of CMS are frequently employed in the breeding of the three-line hybrid *geng* (*O. sativa* ssp. *japonica*). Notably, the D1-type CMS has been a key source for developing ‘Yuza’, ‘Dianza’, ‘Dianheyou’, ‘Chunyou’, and ‘Yongyou’ series hybrid *japonica* or *indica*–*japonica* rice combinations in China [1,2,3].

In 1965, the Chinese scientist Li Zhengyou identified a male sterile (MS) plant from the natural hybridization progeny in a high-altitude *indica* rice landrace and a low-altitude *japonica* rice cultivar ‘Taipei 8’ in Baoshan, Yunnan Province. Following eight generations of backcrossing with the MS plant, the first D1-CMS line was bred in 1969, characterized by gametophytic sterility, and named *japonica* rice ‘Hongmaoying A’ [4,5]. The pollen abortion in D1-CMS occurred at the trinucleate stage, leading to significant starch accumulation, with I_2_-KI staining appearing blue-black. Genetic analysis revealed that the sterile genes (mitochondrial chimeric gene *atp6-orf79*) and restorer genes (*Rf1a* and *Rf1b*) of the D1-CMS were nearly identical to those of the BT-CMS [6,7,8,9]. In 1973, the successful breeding of the first D1-type *japonica* rice R-line 698 marked the establishment of D1-type hybrid rice breeding technology. The first D1-type super-high-yield hybrid rice, ‘Yuza 29’, was a significant breakthrough and set a world record for *japonica* rice yield at 16.63 t/hm^2^ in Binchuan, Yunnan, in 1994 [4]. The synergistic innovation of breeding strategies and germplasm utilization has enabled the effective pyramiding of elite traits. These D1-type varieties of A-, B-, and R-lines, which contain *indica* kinship, not only exhibit rich genetic background and robust multi-resistance but also superior rice quality and high yields, particularly in the southwestern provinces of China. As a result, D1-type hybrid rice holds significant value for functional genomics and genetic improvement in rice [10].

‘DHY615’ is a D1-type three-line hybrid *japonica* rice variety that exhibits high adaptability. It can be cultivated as both irrigated paddy rice and rainfed upland rice under direct seeding, making it well-suited for high-altitude regions up to 2200 m above sea level (masl). This rice variety excels in the diverse ecological zones of the Yunnan–Guizhou (Yungui) Plateau and southwestern China, becoming a predominant variety in these areas. Over the past five years, the total planting area of ‘DHY615’ has exceeded 66,666.67 hectares (hm^2^). Rice cultivation in the southwest region of China, including in the Yungui Plateau and the Qinghai–Tibet Plateau, accounts for approximately 8% of the national rice cultivation area. The region is divided into three sub-areas: the eastern Guizhou and western Hunan Plateau Mountains (single and double cropping rice), the Yunnan–Sichuan Plateau Ridges and Valleys (single cropping rice and double harvest), and the Qinghai–Tibet Plateau cold river valleys (single cropping rice). The accumulated temperature of ≥10 °C ranges from 2900 to 8000 °C, with sunshine hours varying from 800–1500 h and a significant variation in rice distribution based on altitude. In the low-altitude areas, *indica* rice is primarily cultivated, whereas *japonica* rice dominates the high-altitude regions, with a mixed distribution of both rice types in the intermediate zones. The rice grain yield averages between 7.5 and 9.0 t/hm^2^, varying by variety. The rice production season lasts from 180 to 260 days, and annual precipitation ranges from 500 to 1400 mm [11]. This unique plateau *japonica* rice region [12,13] is characterized by its high altitudes, intense solar and ultraviolet (UV) radiation, infertile soil, and frequent natural disasters, necessitating special traits in cultivated rice varieties. These include cold tolerance, strong resistance to UV radiation, efficient fertilizer utilization, and broad adaptability to other harsh environments.

Studies on the structural genomics, functional genomics, and pan-genomics of rice have progressed rapidly in the recent two decades [14]. The applications of molecular markers and SNPs chips have enabled the development of high-yield rice and green super rice varieties [15,16,17]. Several genomic breeding chips have been developed globally for rice, such as Rice6K, Rice60K, Rice90K, and C6AIR [18,19]. Notably, Rice6K incorporates markers of genes such as *GS3* (grain length), *GS5* (grain width), *NRT1.1B* (nutrient-use efficiency), *Badh* (fragrance), *IPA1* (plant architecture), and *Sd-1* (plant height), applying in the genotyping of germplasm resources, gene fingerprint identification of varieties, QTL mapping, marker-assisted backcross breeding, and pedigree verification of breeding lines [20,21]. As quantitative genetics, population genetics, and comparative genomics continue to develop, the genetic mechanisms underlying heterosis in hybrid *indica* rice are becoming increasingly clearer. A few heterosis-associated loci from female parents explained a large proportion of the yield advantage of hybrids over their male parents [22]. The dominance, overdominance, and epistasis underlying functional complementation are indispensable contributors to yield heterosis [23,24]. However, the genotyping and genetic mechanisms underlying the heterosis of D1-type hybrid *japonica* rice remain unclear.

The objectives of this study are to (1) analyze the genealogy and phenotypic characterization of ‘DHY615’, (2) evaluate and validate the aggregation of the excellent agronomic traits in ‘DHY615’ for breeding the plateau hybrid *japonica* rice, and (3) annotate and predict the effects of genetic variation on gene function in ‘DHY615’, and identify genes likely associated with biotic and abiotic stress tolerance (including disease, drought, cold, high iron content in red soils, and high UV radiation), as well as nitrogen use efficiency, productivity, quality, and other important traits related to the wide adaptability of ‘DHY615’. These findings provide a better understanding of the agronomic performance and genetic basis of hybrid *japonica* rice, particularly its adaptability to high-altitude regions, and contribute to agricultural sustainability and food security in diverse rice-growing zones.

## 2. Results

### 2.1. Ecological Adaptation and Field Performance of ‘DHY615’

To validate the effectiveness of the aggregation of the numerous outstanding agronomic traits in ‘DHY615’ for breeding purposes, we compiled data on the distribution and adaptability, grain yield and quality, biotic/abiotic resistance exhibited by ‘DHY615’ throughout its participation in China’s National and Yunnan Provincial Variety Approval and Production Trials. Available online: https://www.ricedata.cn/variety/varis/618086.htm (accessed on 6 November 2024).

These authoritative results indicate that ‘DHY615’ is predominantly distributed and well-adapted across five provinces in China, encompassing the mid-season *japonica* rice region in southwestern China (Yunnan, Guizhou, and Sichuan provinces, and Chongqing Municipality) along the Upper Yangtze River, as well as the *indica*–*japonica* rice mixed cultivation area in Shaanxi Province. Geographically, this spans from 97°79′ E (Dehong Prefecture, Yunnan) to 107°98′ E (Qiannan Prefecture, Guizhou) longitudinally and from 21°52′ N (Mengla Country, Yunnan) to 33°06′ N (Hanzhong Country, Shaanxi) latitudinally, encompassing 10.19 degrees of latitude and 11.54 degrees of longitude, specifically for irrigated paddy fields (Figure 1A).

In addition to irrigated cultivation, ‘DHY615’ is suitable for rainfed conditions in the high-altitude regions of the Yunnan, Guizhou, and Sichuan provinces. Rainfed upland cultivation has expanded to over 60 counties across 14 prefectures/cities (Figure 1B), with altitudes ranging from 826 to 2200 masl in Yunnan Province. The optimal regions for natural rainfed upland planting are characterized by hot and rainy climates, with average altitudes of 800–1600 masl and annual rainfall exceeding 1200 mm. These regions include the Lincang, Honghe, Wenshan, Puer, and Nujiang prefectures. Regions with altitudes between 1600 and 2200 masl, such as Qujing, Yuxi, Dali, Chuxiong, Baoshan, and Kunming, are considered moderately suitable. From 2021 to 2024, ‘DHY615’ exhibited robust growth and a compact plant architecture under direct seeding conditions in rainfed upland fields, as observed in Yunshan Village, Zhutang Township, and Lancang County (Figure 1C). Its development has been recognized as a milestone in the renewal of D1-type hybrid rice, highlighting its exceptional adaptability to high-altitude environments in the Plateau.

In 2013, we conducted varietal comparison trials in Fumin paddy fields, and the results demonstrated significant yield differences among the parental and hybrid F_1_ varieties. ‘DHY615’ achieved the highest average yield at 11.19 t/hm^2^, followed by the maintainer line ‘H479B’ at 8.37 t/hm^2^ and the restorer line ‘Nan 615’ at 7.38 t/hm^2^. ‘DHY615’ exhibited significant heterosis, with yield advantages of 25.1% over ‘H479B’ and 34.1% over ‘Nan 615’.

The yield performance of ‘DHY615’ in paddy fields and rainfed upland planting areas was comparable from 2020 to 2024 (Table 1). Across 22 irrigated paddy field locations (934–2200 masl), the grain yield ranged from 6.97 to 14.27 t/hm^2^, with an average yield of 11.09 t/hm^2^ and a coefficient of variation of 13.00%. Similarly, on-farm demonstration yields in 22 rainfed upland cultivation areas (1075–2132 masl) ranged from 5.64 to 12.22 t/hm^2^, with an average yield of 8.13 t/hm^2^ and a coefficient of variation of 10.43%. The average highest yield of ‘DHY615’ in the most suitable region (Langcang, Puer) reached 9.99 t/hm^2^ under rainfed upland cultivation, showing no significant difference compared to its yield under paddy cultivation. For irrigated paddy fields, among 22 locations, only four (Maguan, Mangshi, Yunxian in Yunnan, and Bijie) showed yields statistically indistinguishable from the lowest-yielding locations (Fuping, Shaanxi). The remaining 17 locations revealed significantly higher yields than the Fuping locations. However, the yield statistical analysis revealed no significant yield differences among the 22 rainfed upland cultivation locations compared to the lowest-yielding location (Anshun, Guizhou), indicating consistent performance across diverse rainfed environments.

According to the Chinese Agricultural Industry Standard NY/T 593-2013 [25], the quality of ‘DHY615’ was evaluated in four irrigated paddy field ecological regions—Lufeng, Yunnan; Lueyang, Shanxi; Huishui, Guizhou; Anshi, Hubei—and in rainfed upland cultivation areas (Yanshang, Yunnan). The results indicated that ‘DHY615’ has a high brown rice percentage (82.50–84.10%) and milled rice percentage (63.00–68.8%), intermediate amylose content (16.4–18.4%), a gel consistency exceeding 64.0 mm (Appendix A), and transparent, polished grains (Appendix A). The grain quality met the II- or III-class standards. Additionally, the grain quality traits of the D1-type hybrid *japonica* ‘DHY34’ (H479A/Nan34) were evaluated under both paddy and rainfed upland field cultivation in Yunnan Province, and it also met the II- or III-class standards. Notably, the grains exhibited enriched micronutrients, including iron (Fe), zinc (Zn), selenium (Se), magnesium (Mg), and manganese (Mn), while remaining free of heavy metals (Appendix A). ‘DHY615’ also showed resistance to multiple major rice diseases, such as leaf blast, spike blast, bacterial blight, and sheath blight, and exhibited high resistance to rice false smut. Furthermore, it demonstrated combined tolerance to cold and drought stress (Appendix A).

In conclusion, DHY615’ demonstrated a clear advantage in yield and quality, along with robust resistance or tolerance to multiple rice diseases, as well as resistance to cold and drought. ‘DHY615’ is distinguished by its exceptional yield potential, superior quality, multi-disease resistance, and adaptability to diverse rainfed upland environments, making it a valuable resource for rice production in challenging environments.

**Table 1 ijms-26-03246-t001:** The yield performance of ‘DHY615’ in paddy fields and rainfed upland planting representative areas.

Paddy Field Locations	Altitude (masl)	Yield (t/hm^2^)	Rainfed Upland Locations	Altitude (masl)	Yield (t/hm^2^)
YN, Lincang, Yunxian	1580	8.02 ± 0.78	YN, Wenshan, Maguan	1260	7.18 ± 0.56
YN, Qujing, Qilin	2000	10.85 ± 1.21 *	YN, Wenshan, Guangnan	1280	7.90 ± 0.25
YN, Qujing, Luliang	1800	12.75 ± 0.45 ***	YN, Kunming, Yiliang	1801	7.20 ± 1.23
YN, Dali, Xiagua	1970	12.15 ± 0.75 ***	YN, Kunming, Xishan	1905	7.56 ± 0.34
YN, Dali, Heqing	2100	11.32 ± 1.66 **	YN, Kunming, Luquan	1900	7.63 ± 0.67
YN, Dali, Yongping	1690	10.73 ± 0.76 *	YN, Kunming, Shilin	1870	7.64 ± 0.99
YN, Dali, Jianchuan	2200	11.50 ± 0.88 **	YN, Nujiang, Fugong	1800	8.78 ± 1.26
YN, Kunming, Xundian	1553	11.64 ± 0.92 **	YN, Nujiang, Lushui	1637	8.97 ± 0.94
YN, Kunming, Yiliang	1540	12.07 ± 1.33 ***	YN, Chuxiong, Yuanmou	1445	7.68 ± 0.92
YN, Wenshan, Maguang	1337	10.28 ± 1.07	YN, Chuxiong, Lufeng	1584	9.09 ± 1.33
YN, Dehong, Mangshi	1516	9.46 ± 1.23	YN, Lincang, Linxiang	1813	7.91 ± 0.45
YN, Chuxiong, Lufeng	1548	12.79 ± 1.33 ***	YN, Lincang, Yunxian	1580	8.77 ± 0.72
YN, Hehong, Jianshui	1700	12.27 ± 0.56 ***	YN, Honghe, Hekou	1075	7.71 ± 1.35
YN, Lijiang	2195	10.84 ± 2.1 *	YN, Honghe, Luchun	1306	7.51 ± 2.55
YN, Yuxi, Eshan	1845	11.84 ± 0.12 ***	YN, Honghe, Yuanyang	1700	7.69 ± 1.54
YN, Zhaotong, Ludian	1850	12.23 ± 2.04 ***	YN, Honghe, Kaiyuan	1450	9.65 ± 1.72
YN, Baoshan, Longyang	1650	12.55 ± 0.33 ***	YN, Honghe, Gejiu	1499	8.93 ± 3.29
GZ, Bijie	1520	8.93 ± 0.54	YN, Puer, Mojiang	1700	7.53 ± 0.36
GZ, Qiannan, Huish	1400	11.13 ± 1.95 **	YN, Puer, Ninger	1600	9.20 ± 1.79
SC, Liangshan, Xichang	1480	10.86 ± 1.34 *	YN, Puer, Lancang	1460	9.99 ± 1.81
SC, Liangshan, Mianning	1620	12.39 ± 0.76 ***	GZ, Anshun	2132	7.01 ± 0.38
SX, Weinan, Fuping	934	7.47 ± 0.50	SC, Jingchuan	1300	7.37 ± 1.47
Mean		11.09			8.13
SD		1.44			0.85
Coefficient variation (%)		13.05			10.43

Note: YN: Yunnan Province, GZ: Guizhou Province, SC: Sichuan Province, SX: Shaanxi Province locations (province, prefecture or county, and city). The data was compiled derived from the National/Provincial field yield trials and on-farm demonstrations in 2020–2024. The yield measurement was conducted in accordance with the Chinese Agricultural Industry Standard NY/T 789-2004 [26]. *: yield given in average ± SD. The asterisks indicate significant differences: * *p* < 0.05, ** *p* < 0.01, and *** *p* < 0.001. Grain moisture derived using a PM-8188-A grain moisture meter.

### 2.2. Genealogy and Phenotypic Characterization of ‘DHY615’

‘DHY615’ was developed from the female parent (A-line), ‘H479A’, and the male parent (R-line), ‘Nan615’. Its genealogy (Figure 2) reveals a rich and diverse genetic heritage, integrating Yunnan local rice resources with elite genetic materials from *indica* and *japonica* rice subspecies and ecotypes across Asia (e.g., China, Japan, Korea, and Philippines–IRRI), Europe (Italy), and America (USA)). This highlights the successful application of advanced breeding technologies in developing D1-CMS lines and hybrid *japonica* rice in China.

The morphology of ‘DHY615’ (Appendix A) was conducted alongside evaluations of yield-related agronomic traits with ‘H479A’ and ‘Nan615’. The results showed that ‘DHY615’ exhibits significantly greater plant height, effective spike number, and grain number per panicle compared to ‘Nan615’. Its effective spike number also surpassed that of ‘H479A’. These findings indicate that ‘DHY615’ inherits dominant traits from both parents, including enhanced spike productivity, grain number per panicle, moderate plant height, appropriate thousand-seed weight, and a stable seed-setting rate. The total growth duration of ‘DHY615’ was approximately 150 days under rainfed upland conditions and 180 days in paddy fields, varying with altitude (Appendix A).

### 2.3. Indica–Japonica Component and Genomic Differences

The SNP profiles of ‘DHY615’ and its parental lines were compared with the reference genomes of ‘Nipponbare’ and ‘9311’. Chromosomal analyses revealed that ‘H479A’ (Figure 3A), which carries the *indica* variety ‘Lianyuanzao’ in its pedigree, exhibited an *indica* component of 1.47% and a *japonica* component of 94.13%, aligning most closely with the ‘Nipponbare’ genome. In contrast, ‘Nan615’ (Figure 3B), an *indica*–*japonica* bridging R-line, displayed an *indica* component of 16.27% and a *japonica* component of 74.27%. The genomic composition of ‘DHY615’ (Figure 3C) was intermediary, with 87.60% *japonica* and 3.60% *indica* components.

Analysis using the rice GSR40k SNPs chip identified 32,887 high-quality SNPs, of which 32,607 were mapped to specific genomic coordinates. These loci were distributed across the 12 rice chromosomes (Figure 3D–F). ‘H479A’ had five heterozygous SNPs loci, with a homozygosity rate of 99.98% (Figure 3D), and ‘Nan615’ exhibited 24 heterozygous loci, with a homozygosity rate of 99.93% (Figure 3E). Both parental lines displayed stable genotypes, without phenotypic segregation. Conversely, the homozygosity rate of ‘DHY615’ was 80.19%, with 6432 heterozygous SNPs (Figure 3F). These loci were unevenly distributed, with chromosome 1 containing the most heterozygous loci and chromosome 5 containing the fewest heterozygous loci (Figure 4A). A total of 25,629 homozygous SNPs loci (78.60% of all SNPs markers) were identical across ‘DHY615’ and its parents, while 19.72% of markers were heterozygous, indicating that the heterozygous advantage of ‘DHY615’ likely resides in these genomic regions. The detailed analysis of the 6432 heterozygous loci (Figure 4B) showed that 57.48% were in promoter regions, 22.06% in introns, 2.83% in 3’UTRs, 3.20% in 5’UTRs, and 14.43% in coding sequences.

### 2.4. Functional Impact of SNPs Variations

The functional effects of SNPs variations were predicted using the ANNOVAR program (ANNOVAR version 2023-06-08). Variants were classified into four impact categories: high, moderate, low, and modifier. The chromosomal distribution of variant effects across the four groups revealed significant disparities (Figure 5A), particularly with the number of heterozygous loci in ‘DHY615’. Chromosome 1 exhibited the highest variation density, whereas chromosome 9 had the lowest. The high-impact group, consisting of 218 variants (0.35%), included splice–site, stop–gain, stop–loss, and start–loss mutations. Moderate-impact variants (3488, 5.63%) were predominantly missense mutations. Low-impact variants (1791, 2.89%) comprised synonymous and splice–region changes. The majority (91.12%) were modifier variants (56,403), encompassing intron and intergenic variants. The stop–gain mutations dominated the high-impact group, while missense variants were the most common in the moderate-impact group (Figure 5B).

Additional details on high- and moderate-impact variants are provided in Appendix A. Among 64 high-impact variants referenced in the RAP-DB database. Available online: https://rapdb.dna.affrc.go.jp (accessed on 6 November 2024) (Appendix A), genes such as *OsLPR4*, *OsSCP42*, *OsGELP26*, and *OsWAK47* were likely implicated in their biochemical processes, seed development, and morphological traits. Stress-response genes like *OsSPX6*, *OsNCED2*, and *PR10* were also identified, along with the resistance genes Pi3, *OsWRKY123*, *OsLOX11*, and *Bph14.*

Of the 428 heterozygous SNPs loci in the coding regions, 80 were identified as high- and moderate-impact variants (Appendix A). Genes including *OsMC6*, *OsWRKY22*, and *OsACBP3* were likely involved in both abiotic and biotic stress responses. Additional genes, such as *OsPUB58*, *OsMYB1R1*, and *OsSPX1,* were likely linked to tolerance against salt, cold, and drought stresses (Figure 6). Disease-resistance genes like *Pi64*, *CYP76M6*, *OsCIPK05*, and *OsBSR-K1* were implicated in resistance to leaf blight, insect infestation, and blast disease (Table 2). These findings suggest that most of these genes may have contributed to the enhanced stress and disease resistance observed in ‘DHY615’.

### 2.5. Functional Locus Analysis of ‘H479A’, ‘Nan615’, and ‘DHY615’

We identified key genes likely associated with essential agronomic traits in ‘H479A’, ‘Nan615’, and ‘DHY615. The results reveal that ‘DHY615’ inherits alleles from ‘Nan615,’ including *SKC1*, which enhances root sodium content for improved salt tolerance [37]; *qUVR-10*, which increases UV–B tolerance [38]; *NRT1.1B*, associated with enhanced nitrogen use efficiency; *STV11*, conferring resistance to rice stripe virus; *OsAAP6*, promoting high protein content; *GS3*, increasing stigma exertion rate and potentially grain size [39]; *Hd17*, delaying heading and flowering [40]; and *Os11g08410*, also delaying heading. Notably, alleles shared by both parents, such as *Waxy* and *ALK*, responsible for taste quality [41,42]; *Hd3a* and *Os01g62780*, controlling heading date [43]; *SaF*, conferring wide *japonica* affinity [44]; *sh4*, associated with the seed shattering trait [45]; *OsFRO1*, imparting iron tolerance [46]; *Rymv1*, associated with resistance to yellow mottle virus disease [47]; and *Sdt97*, contributing to semi-dwarfism [48], were also inherited by ‘DHY615. However, genes such as *Sd-1* and *qSW5/GW5* from ‘Nan615’ were lost during breeding, while *SUB1A* was newly acquired in ‘DHY615’. These findings indicate that ‘Nan615’, as the R-line, carries more functional alleles than the A-line ‘H479A’. Consequently, ‘DHY615’ exhibits trait complementarity and promotes favorable phenotypes (Table 3).

To validate the functional loci, gene amplification analysis of *Rf1a* and *OsBADH2* was conducted in ‘DHY615’ and its parents (Appendix A). The results confirm that the R-line ‘Nan615’ carries the *Rf1a* gene, while the A-line ‘H479A’, B-line ‘H479B’, and hybrid ‘DHY615’ harbor the *OsBADH2* gene. These findings affirm that the aggregation of functional genes in ‘DHY615’ positively impacts the breeding of D1-type *japonica* hybrid rice for practical production.

## 3. Discussion

The variety ‘DHY615’ represents the culmination of 50 years of D1-type *japonica* hybrid rice breeding. Possessing a highly diverse genetic background, ‘DHY615’ demonstrates exceptional ecological adaptability to the high-altitude regions of the Yungui Plateau in southwestern China (Figure 1 and Figure 6). Its successful cultivation exemplifies the breeding value and molecular mechanisms underlying plateau *japonica* hybrids, serving as a vital reference for future research and development. The genealogical tracing of the D1-type hybrid *japonica* rice, ‘DHY615’ (Figure 2), reveals a rich integration of superior *indica* and *japonica* genetic resources from both domestic and international origins [4]. This lineage was achieved by creating an R-line through an “*indica*–*japonica* bridging” strategy, particularly involving the crossbreeding of IR8 and Keqing No.3. This approach facilitated extensive genetic exchange between subspecies. We analyzed the genetic polymorphism and structure of *japonica* rice populations in the Yungui Plateau, showing that *japonica* rice resources predominantly consist of the ‘Dianza’, ‘Dianheyou’ (DHY), and ‘Hexi’ series [49]. The genetic composition of ‘Nan615’ and ‘DHY615’ reflects a high proportion of *indica* components (Figure 4), demonstrating the historical application of a three-line breeding strategy for D1-type *japonica* hybrids through *indica*–*japonica* integration.

In this study, by utilizing high-impact SNP variants, we identified hundreds of functional genes in ‘DHY615’ and its parental lines. These encompass known genes involved in plant architecture, grain yield, quality, and resistance to both biotic and abiotic stresses, such as *NRT1.1B*, *SKC1*, *qUVR-10*, *GS3*, *Waxy*, *ALK*, *Hd3a*, and *OsFRO1* (Table 3). These genes may have contributed to key traits, including adaptability to cold, drought, and poor soil conditions, as well as resistance to high UV and iron levels (Figure 1 and Figure 6). This underscores the molecular and genetic basis of ‘DHY615’’s high-altitude adaptability in the Yungui Plateau.

We further identified 32,607 SNPs in ‘DHY615’ and its parents, of which 6432 were completely heterozygous markers. This extensive variation likely underpins the superior phenotypic traits of ‘DHY615’, including improved plant architecture, disease resistance, grain quality, and tolerance to environmental stresses. Using the ANNOVAR program, we predicted the functional impacts of these variants. Among them, 218 variants (0.35%) were classified as high impact, involving changes such as stop codon gain or loss, start codon loss, and splice–site alterations. These variants are strongly associated with phenotypic variation [50]. Additionally, 53 high-impact SNPs were annotated to agronomically important genes, while 3488 (5.63%) missense mutations with moderate functional effects may also influence phenotypic traits. Even variants predicted to have low or modifier effects could contribute to observed phenotypic diversity, as evidenced by the 428 heterozygous SNP loci in the coding regions predicted to have these impacts. Thus, ‘DHY615’ likely exhibited complementarity in key traits and promoted favorable phenotypes. It was particularly significant that scholars confirmed through field experiments that *japonica* varieties carrying the *NRT1.1B indica* allele showed a substantial increase in seed yield and nitrogen use efficiency compared to those without the *NRT1.1B* allele [51], and the genotyping of ‘Nan615’ for *NRT1.1B* was consistent with the representative variety. The findings of this study indicated that ‘Nan615’ also harbors more significant genes, potentially conferring desirable traits, which align with the outcomes previously reported [22]. Consequently, the hybrid loci contributing to the heterotic advantage are primarily derived from ‘Nan615’. Building on this, selecting beneficial traits from the parent ‘H479A’ could be harnessed to further enhance rice quality. The fragrance gene from ‘DHY34’ also inherits alleles from ‘H479A’ [52]. Coupled with the results of the fragrance gene detection, we can affirm the reliability of functional gene analysis. Furthermore, we tested the field drought resistance of 198 *japonica* rice variety resources [53], demonstrating that ‘DHY615’ exhibited high resistance to drought (Appendix A). This is consistent with the potential of ‘DHY615’ to be cultivated as upland rice in high-altitude areas under rainfed conditions.

The ecological adaptation and field performance of ‘DHY615’ were assessed by collecting phenotypic data related to on-farm yield, disease resistance, and grain quality (Table 1 and Appendix A). Field trials confirmed the yield stability of ‘DHY615’ under rainfed cultivation across diverse environments in Yunnan, Guizhou, and other regions in southwest China, with no statistically significant yield difference between the optimal and marginal growing zones (Figure 6). Furthermore, the yield variability observed in irrigated locations could potentially be attributable to different management practices and environmental pressures. Additionally, ‘DHY615’’s capability for rainfed upland cultivation was compared with other water-saving and drought-resistant rice (WDR) varieties. Li et al. analyzed the yield of 12 approved varieties in Hubei Province and found that the yield of current WDR varieties was 7.5 t/hm^2^~9.8 t/hm^2^, still lack significant advantages in yield compared to irrigated paddy field rice varieties and ‘DHY615’ (rainfed upland planting) [54]. However, the yield performance of ‘DHY615’ in paddy fields and rainfed upland planting areas was comparable, suggesting robust eco-adaptation, particularly valuable for climate-resilient cultivation in montane areas.

Notably, ‘DHY615’ could be effectively utilized as a rainfed upland rice variety (Figure 6), showcasing suitable ecological characteristics for water-saving and drought-resistant rice (WDR) [55]. It has transformed traditional seedling raising, plowing, and transplanting practices, breaking away from the conventional paddy rice cultivation that yielded low productivity, high cost, and inefficient use of resources (and, in some instances, labor). The success of ‘DHY615’ has laid the foundation for further innovation in sustainable ecological agriculture. Its cultivation alongside other crops offers opportunities to optimize land use and minimize pest and disease pressure. In this study, we have utilized the GSR40K SNP chip for a preliminary analysis of the genetic background and broad adaptability of ‘DHY615’. Our ongoing research aims to integrate the haplotype-resolved telomere-to-telomere (T2T) and high-throughput chromosome conformation capture (Hi-C) reference genome, multi-omics, pan-omics, and advanced biotechnology for the design and genomic selection (GS) of DHY hybrid *japonica* rice. Furthermore, we are conducting a long-term environmental impact assessment of D1-type hybrid upland rice cultivation, focusing on water and land use efficiency and environmentally sustainable agriculture, which includes reducing CH_4_ gas emissions, total nutrient runoff loss, and total pesticide loss. Future breeding efforts will aim to expand the DHY series of hybrid *japonica* rice varieties, ensuring their suitability for rainfed upland agriculture while maintaining high yield, quality, and resilience.

## 4. Materials and Methods

### 4.1. Research Materials

The rice variety ‘DHY615’, approved by both the Crop Variety Committee of Yunnan (approval No.2017017) and China (approval No. CHN20220301), is an exceptional D1-type three-line *japonica* hybrid rice combination. It is characterized by the CMS line (A-line) ‘H479A’ as the female parent and the restorer line (R-line) ‘Nan615’ as the male parent. In 2019, this variety was bestowed with the Gold Award in the Japonica Rice Category at the 2nd National Quality Rice Tasting Evaluation. In 2023, ‘DHY615’ received the “Breakthrough New Rice Variety” accolade from the Department of Agriculture and Rural Affairs of Yunnan. All rice cultivars used in this study were developed by the Rice Research Institute of Yunnan Agricultural University.

### 4.2. Breeding of ‘DHY615’

The maintainer line (B-line) ‘H479B’ was developed by another culture of F_1_ (Dianyu No.1/Dianxun No.8//Lianyuanzao), and the D1-CMS A-line ‘H479A’ was developed by the continuous backcrossing of ‘H479B’ with ‘Hexi 42A’. Therefore, ‘H479A’ inherited the high-yield and ideal plant type from ‘Dianyu No.1’, as well as its good taste and fragrance from ‘Dianxun No.8’. The R-line ‘Nan615’ was bred by crossbreeding, using an elite core R-line ‘Nan34’ as the female parent, with ‘Ansanbyeo’ introduced from South Korea as the male parent for hybridization to obtain F_1_ in 2003. From 2003 to 2009, continuous self-crossed breeding and hybrid offspring selection were conducted in Kunming and Hainan based on dominant agronomic characteristics. Until 2010, the R-line ‘Nan615’ was self-pollinated until the F_13_ generation, with the dominant agronomic traits being completely stable and uniform.

### 4.3. Phenotyping and Assessment of Field Performance

Assessment of the phenotypic and field performance of ‘DHY615’ under both paddy and upland cultivation of ‘DHY615’ was carried out in areas below 2200 masl (Yunnan *japonica* growing region) that are suitable for upland rice cultivation from 2020 to 2024. The agronomic practices for ‘DHY615’ were implemented in accordance with the Technical Specification for Cultivation of Upland High-quality Rice (DB53/T 1318.1-2024) [56] of the Yunnan Province Standard, including seed treatment, irrigation methods, fertilization, pest management, planting density, etc. The comprehensive characteristics of ‘DHY615’, including the varietal feature of DUS (distinctiveness, uniformity, and stability), biological traits (plant height, effective spikes, panicle length, grain number per panicle, seed-setting rate, thousand seed weight, etc.), yield, quality, disease resistance, cold tolerance, drought tolerance, and ecological adaptability, were evaluated according to the National and Provincial field trial standards (DB42/T 1404-2018 and NY/T 2863-2015) [57,58]. The grain yield of ‘DHY615’ in more than 40 core on-farm demonstration locations was measured in the field by experts organized by National or Provincial Science and Technology Departments, following the “Rice Yield Measurement Method” of the Ministry of Agriculture and Rural Affairs of China.

### 4.4. DNA Extraction and High-Density Genome-Wide SNP Microarray Analysis

‘H479A’, ‘H479B’, ‘Nan615’, and ‘DHY615’ (F_1_) were grown in a greenhouse at the Rice Research Institute of Yunnan Agricultural University. The fresh young seedling leaflets of 3–5 cm in length were harvested and ground, after freezing with liquid nitrogen, and the genomic DNA of ‘H479A’, ‘H479B’, ‘Nan615’, and ‘DHY615’ (F_1_) was extracted following the CTAB protocol. The DNA quality was assessed by 1–1.5% agarose gel electrophoresis. The DNA samples with high quality (>10-kb fragments) and an appropriate concentration (10–50 ng/uL) were analyzed using the whole genome SNP array GSR40K containing 44263 variations, which was developed by Wuhan Shuanglvyuan Chuangxin Technology Research Institute Co. LTD, Wuhan, Hubei Province). The overall steps involved DNA amplification, fragmentation, chip hybridization, single base extension, staining, and scanning, all of which were performed according to the Infinium assay standard protocol (Infinium HD Assay Ultra Protocol Guide. Available online: http://www.illumina.com/ (accessed on 6 November 2024)). A HiScan scanner (Illumina Inc., San Diego, CA, USA) was used for the SNP microarray scanning, and GenomeStudio software version v2.0.5 was used for raw SNP genotyping analysis. The SNP markers were filtered using the following criteria: (1) a call rate ≥80% (missing data < 20%), and (2) a MAF (minor allele frequency) ≥5%. The resulting high-quality SNPs were used for downstream analyses. The R platform (R version 4.4.2) was employed for genotype identification, comparison, and map drawing [59]. The unique chromosome frames were constructed based on the ‘Nipponbare’ 7.0 Genome. Based on the genomic sequence of *japonica* ‘Nipponbare’, *indica* ‘9311’ was used to analyze the *indica–japonica* components and homozygosity of ‘DHY615’ and its parents. Here, the SNP genotypes of ‘DHY615’ and its two parents were compared and classified as homozygous and heterozygous (male parent genotype AA, maternal genotype BB, and hybrid genotype AB).

Based on the gene sequences of *OsBADH2* and *Rf1a* in ‘Nipponbare’, three pairs of primers were designed in Appendix A. The DNA from the A-line ‘H479A’, R-line ‘Nan615’, B-line ‘H479B’, and hybrids of ‘DHY615’ were used as templates, and the target genes were amplified by Taq DNA polymerase (Thermo Fisher Scientific, Waltham, MA, USA). The total 15 μL PCR reaction system included 0.3 μL of template DNA, 1 μL of primers, and 13.7 μL of mix. The amplification reaction conditions were as follows: pre-denaturation at 94 °C for 5 min, 94 °C denaturation for 30 s, 55 °C/56 °C annealing for 10 s, 72 °C extension for 30 s, 35 cycles, and a final extension at 72 °C for 7 min. Finally, the PCR products were electrophoresed on 1% agarose gel.

### 4.5. Functional Locus and Statistical Analysis

The reference genome IRGSP-1.0 and the annotation of genomes with GFF files from the Rice Genome Annotation Project. Available online: https://rice.uga.edu/ (accessed on 6 November 2024) were used to annotate and extract the genotypes, positions of varieties, and variants with high, moderate, low, and modifier impact effects using ANNOVAR software with default parameters [60]. To identify SNP variations in well-known, agronomically important genes, we referred to the genes listed in the RAP database RAP-DB. Available online: https://rapdb.dna.affrc.go.jp (accessed on 6 November 2024) [61].

The differences in the yield and yield-related characteristics of ‘DHY615’ were analyzed using GraphPad Prism 8.4.2. The average, standard deviation and coefficient of the variations were calculated. The graphs were created using Adobe Photoshop CC 2019 and Adobe Illustrator CC 2017.

## 5. Conclusions

In this study, we conducted a comprehensive analysis of the genealogy and phenotypic characterization of ‘DHY615’, and identified genes likely associated with biotic and abiotic stress tolerance based on genomic annotation and predicting the effects of genetic variation on gene function in ‘DHY615’. Our findings provide valuable insights into the genetic underpinnings of the elite traits and agronomic performance of the D1-type *japonica* hybrid rice, particularly its exceptional adaptability to the diverse high-altitude environments of the Yungui Plateau in southwestern China. We should prioritize the application of ‘DHY615’ in the most suitable regions of five prefectures/cities, while the suboptimal cultivation conditions will require complementary water regulation technologies to achieve the yield potential. Moreover, the yield demonstrated stable performance across diverse rainfed environments, showcasing its stress-resilient yield stability. It also offers valuable resources for developing breeding strategies for marker-assisted selection (MAS) and genomic selection (GS), presenting new opportunities for breeding and unlocking molecular mechanisms in plateau rice varieties. This work lays a foundation for the continued improvements in rice varieties suited to high-altitude environments and highlights the broader potential of ‘DHY615’ in promoting sustainable agricultural practices. The large-scale adoption of ‘DHY615’ via rainfed upland cultivation systems demonstrates significant potential for advancing research in water-efficient agriculture, optimized land utilization, and ecological sustainability in high-altitude plateau regions.

## Figures and Tables

**Figure 1 ijms-26-03246-f001:**
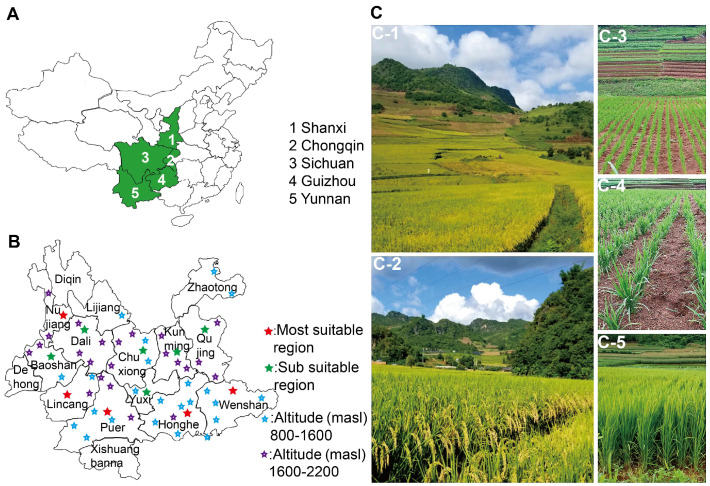
The geographic distribution of the ‘DHY615’ plant area and the growth periods of ‘DHY615’ under rainfed upland conditions. (**A**) The provinces where ‘DHY615’ is significantly planted in southwestern China are indicated in green. (**B**) Considering altitude and annual rainfall, ‘DHY615’ has been found suitable for cultivation under rainfed (non-irrigated direct seeding) conditions across 13 prefectures (encompassing 52 counties) in Yunnan Province. Five prefectures/cities (Nujiang, Lincang, Puer, Honghe, and Wenshan) have been identified as the most suitable cultivation zones (★), while six prefectures/cities (Dali, Chuxiong, Yuxi, Kunming, Qujing, and Baoshan) have been classified as sub-optimal zones (★). (**C**) The various growth stages of ‘DHY615’ in the rainfed upland fields in Lancang County (altitude: 1300–1700 masl, annual rainfall: 1600 mm) in Yunnan Province are as follows: (**C-1**,**C-2**) represent the maturity stage; (**C-3**) is the seedling stage; (**C-4**) is the tillering stage; and (**C-5**) is the heading stage.

**Figure 2 ijms-26-03246-f002:**
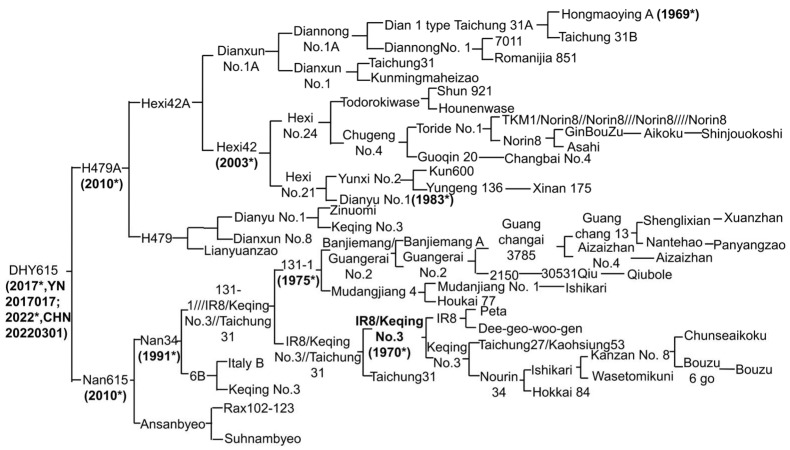
Genealogy of the D1-type hybrid *japonica* rice, ‘DHY615’. * Indicates the year the variety was developed. Over the course of approximately five decades, from 1969 to 2017, through systematic selection and breeding, two elite lines, ‘Nan 615’ and ‘H479A’, were successfully developed. The R-line, ‘Nan615’, was bred by crossing ‘Nan34’ (which includes the *indica* ‘IR8’ and a series of Japanese *japonica* kinships) with Ansanbyeo from Korea. The D1-type CMS line, ‘H479A’, was developed by crossing ‘Hongmaoying A’ with plateau-adapted B-lines (‘Diannong No.1’, ‘Dianxun No.1’, ‘Dianyu No.1’, ‘Hexi42’, and ‘H479’).

**Figure 3 ijms-26-03246-f003:**
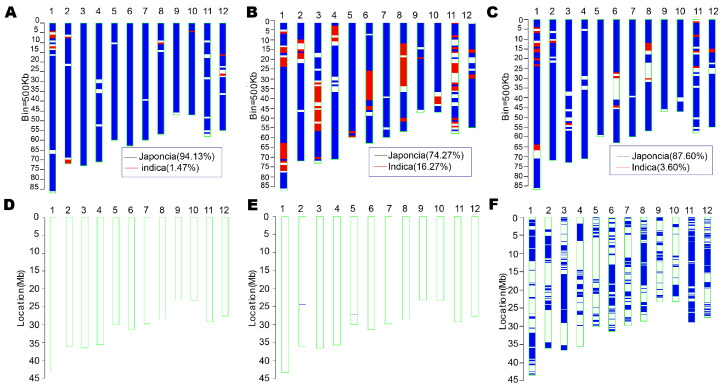
Genetic background screening using the GSR40K array. (**A**) The *indica–japonica* genetic background of ‘H479A’. (**B**) The *indica–japonica* genetic background of ‘Nan615’. (**C**) The *indica–japonica* genetic background of ‘DHY615’, with the blue color representing the *japonica* component and the red color indicating the *indica* component. (**D**) The detailed genotyping map of the plant ‘H479A’. (**E**) The detailed genotyping map of the plant ‘Nan615’. (**F**) The detailed genotyping map of the plant ‘DHY615’. The twelve chromosomes of rice are labeled from 1 to 12. The reference genome is Nipponbare (rice MSU7.0). The blue short line (**D**–**F**) on each chromosome indicates the position of the single nucleotide polymorphisms (SNPs) with heterozygous genotypes, where genomic fragments of the parents (‘H479A’ and ‘Nan615’) were introgressed, and the genotypes of the remaining genomic regions were the same as the parent.

**Figure 4 ijms-26-03246-f004:**
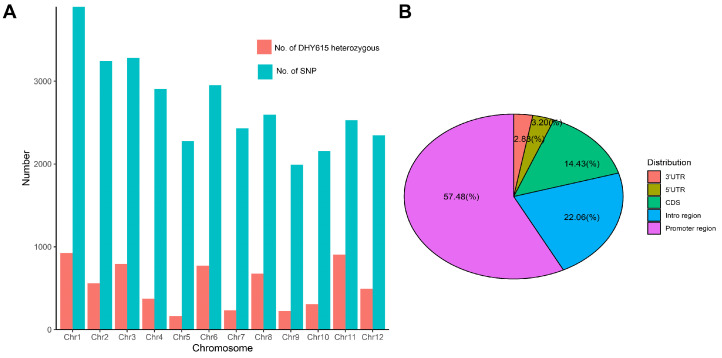
Number and distribution of the heterozygous loci in ‘DHY615’. (**A**) The number of SNPs and heterozygous loci across 12 chromosomes. (**B**) The differential distribution proportions of the heterozygous loci for ‘DHY615’ within the gene structure.

**Figure 5 ijms-26-03246-f005:**
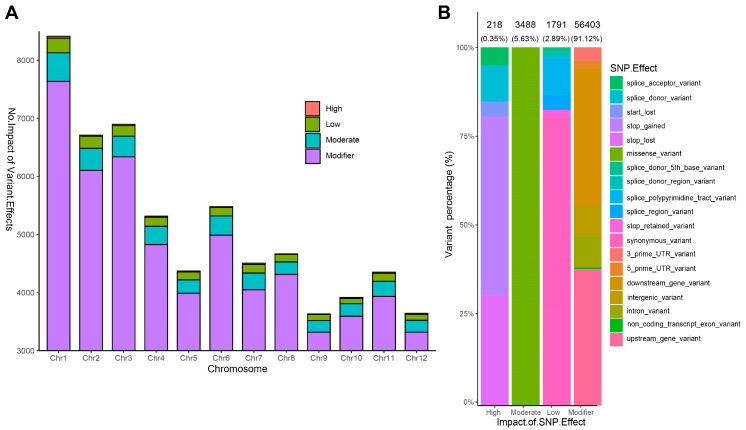
Classification of the variants by their impact on gene function and by the effects of these impacts. (**A**) The distribution of the variant impact effects (high, low, moderate, and modifier group) across the 12 chromosomes. (**B**) The classification of the variants based on the impact effects on gene function.

**Figure 6 ijms-26-03246-f006:**
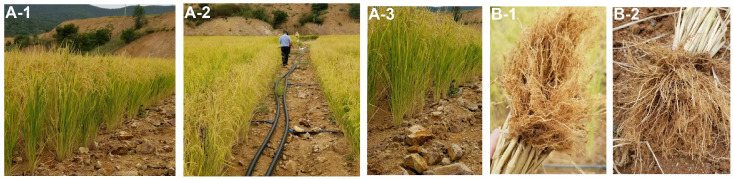
‘DHY615’ water-saving planting in the arid region with poor soil of Yuanmou County (1445 msal, annual rainfall less than 300 mm) in Yunnan (2021). At the maturity stage, ‘DHY615’ exhibits drought-adaptive features (**A-1**–**A-3**). The root structure and density of ‘DHY615’ (**B-1**,**B-2**).

**Table 2 ijms-26-03246-t002:** Summary of SNPs loci variations in abiotic stress and biotic stress genes.

Gene Name	Gene ID	Trait	No. of Variation Sites	Reference
*OsMC6*	LOC_Os01g58580	abiotic stress and biotic stress	1	[27]
*OsWRKY22*	LOC_Os01g60490	abiotic stress and biotic stress	1	[28]
*OsACBP3*	LOC_Os03g37960	cold tolerance and blast disease	1	[29]
*OsPUB58*	LOC_Os10g40120	drought and cold tolerance	1	[30]
*OsMYB1R1*	LOC_Os04g49450	drought resistance	1	[31]
*OsSPX1*	LOC_Os06g40120	cold tolerance	1	[32]
*Pi64*	LOC_Os01g57280	leaf blast disease resistance	1	[33]
*CYP76M6*	LOC_Os02g36280	insect resistance	1	[34]
*OsCIPK05*	LOC_Os01g10890	blast disease resistance	1	[35]
*OsBSR-K1*	Os10g0548200	blast disease	1	[36]

**Table 3 ijms-26-03246-t003:** Summary of genes associated with important agronomic traits in ‘H479A’, ‘Nan615’, and ‘DHY615’.

Trait	Gene	Probe Type	Chr	Representative Variety	Phenotype	H479A	Nan615	DHY615
TQ	*BADH2 (fgr)*	SNP	8	Suyunuo	Fragrance	√		√o
ABS	*SKC1*	SNP	1	Nona Bokra	Salt resistant		√	√o
ABS	*qUVR-10*	SNP	10	Sasanishiki	Enhanced phytoremediation activity		√	√o
ABS	*NTR1.1B*	SNP	10	9311	Enhanced nitrogen uptake		√	√o
BS	*STV11*	SNP	11	Kasalath	Persistent resistance to rice stripe virus		√	√o
TQ	*OsAAP6*	INDEL	1	9311	High protein		√	√o
TQ	*GS3*	SNP	3	Minghui 63	Long grain		√	√o
HD	*Hd17*	SNP	6	Koshihikari	Delayed flowering		√	√o
HD	*Os11g08410*	SNP	11	Haplotype C	Delayed heading		√	√o
TQ	Waxy	SNP	6	Nipponbare	Increase the branched starch content	√	√	√
TQ	*ALK*	SNP	6	Minghui 63	Long branched chain starch and higher pasting temperature	√	√	√
HD	*Os01g62780*	SNP	1	Haplotype B	Delayed heading	√	√	√
HD	*Hd3a*	SNP	6	Nipponbare	Photoperiod-sensitive genes	√	√	√
FE	*SaF*	SNP	1	Nipponbare	Wide affinity	√	√	√
OT	*sh4*	SNP	4	Nipponbare	Non-shattering	√	√	√
ABS	*OsFRO1*	SNP	4	KDML105	Iron tolerance	√	√	√
BS	*Rymv1*	SNP	4	Nipponbare	Resistance yellow mottle virus disease	√	√	√
PA	*Sdt97*	SNP	6	Y98149	Semidwarf	√	√	√
PA	*Sd-1*	INDEL	1	DGWG-type	Semidwarf		√	
TQ	*qSW5/GW5*	INDEL	5	Nipponbare	Increase grain width		√	
ABS	*SUB1A*	SNP	9	FR13A	Flood-resistant	o	o	√o

Note: The “√” symbol indicates the genotypic consistent with the representative variety and does not signify whether mutations have occurred; “√o” indicates heterozygous; the “o” symbol indicates the material has no allele. The genes highlighted with yellow and blue are inherited by ‘DHY615’ from ‘H479A’ and ‘Nan615’, respectively. The genes highlighted with green are inherited by ‘DHY615’ from both ‘H479A’ and ‘Nan615’. The unhighlighted genes are unique to ‘DHY615’ and ‘Nan615’, respectively. HD, heading date; FE, fertility; TQ, taste quality; OT, others; ABS, abiotic stress; BS, biotic stress; PA, plant architecture.

## Data Availability

The data presented in this study are available upon request from the first author, Chunli Wang. Due to privacy restrictions, the data are not publicly available.

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
