# Peer review of "Breeding D1-Type Hybrid Japonica Rice in Diverse Upland Rainfed Environments"

_ijms, 2025, doi:10.3390/ijms26073246_

Round 1

Reviewer 1 Report

Comments and Suggestions for Authors

The abstract is well-structured, but it would benefit from a clearer statement on how this study differs from previous work on D1-type hybrid japonica rice.

The introduction provides historical context but should better connect past research to the present study’s objectives.

The discussion section should explicitly relate the study's findings to real-world agricultural practices and climate resilience.

Some key terms, such as "heterozygous SNPs" and "functional allele genes," should be briefly explained for a broader audience.

The manuscript would benefit from a clearer methodological explanation of how SNP variants were identified and classified.

The conclusion could be expanded to include specific recommendations for future research.

Line 21-22: The phrase "thousands of heterozygous SNPs" is vague. Can you provide a precise number or a more refined estimate?

Line 25-26: The mention of "high UV radiation" as a stress factor is interesting. Can you provide references or specific data supporting this?

Line 30-31: The sentence “underscoring its potential for sustainable rice production” should include specific evidence from the study to support this claim.

Line 36-38: The introduction states that WA and HL are primary CMS sources, but how does D1 compare in terms of practical advantages or disadvantages?

Line 43-44: Consider rewording "a male sterile (MS) plant from the natural hybridization progeny" to improve clarity and readability.

Line 47-48: The description of the pollen abortion process could be made clearer—perhaps with an additional figure or illustration.

Line 49-50: The mention of atp6-orf79 and Rf1 genes is crucial; a short explanation of their significance for non-specialists would be helpful.

Line 51: The phrase “marked the establishment of D1-type hybrid rice breeding technology” should be revised to clarify the significance of this development.

Line 325-326: Clarify the term “D1-type three-line japonica hybrid rice combination.” Does "D1-type" refer to a specific classification system? Adding a reference would be helpful .

Line 327-328: The sentence "In 2019, this variety was bestowed with the Gold Award..." lacks citation. Consider providing a reference or further explanation regarding the awarding institution.

Line 335-336: The explanation of ‘Nan615’ development could benefit from a more detailed description of its breeding process, including key genetic traits introduced from ‘Ansanbyeo’ .

Line 343-344: The phrase "several diversified regions with different elevations" is vague. It would be beneficial to specify the range of elevations tested to improve clarity.

Line 347-348: Provide additional details or references for the field trial standards (DB42/T 1404-2018 and NY/T 2863-2015). If these are widely recognized standards, a brief description of their scope would be useful.

Line 349-350: The term “core on-farm demonstration locations” should be defined more precisely. What criteria were used to select these locations? Are they representative of typical rice-growing areas?

Line 420-421: The funding section mentions grants, but it would be helpful to briefly specify how these funds contributed to the study (e.g., data collection, experimental design, equipment).

Line 426: The "Data Availability Statement" states that all data is contained within the article. However, the manuscript references supplementary tables (e.g., S1-S7). Consider specifying whether raw datasets are available elsewhere.

Line 417-419: In the “Author Contributions” section, clarify the specific roles of each author, especially regarding supervision and conceptualization. Were there any significant contributions from external collaborators?

Line 429: The acknowledgment section lists names but does not mention specific contributions. Were these individuals involved in data collection, statistical analysis, or manuscript editing? Providing details would give better credit to their contributions.

Comments on the Quality of English Language

Throughout the manuscript, there are several instances of awkward phrasing and grammatical errors that detract from the overall clarity. For example, in lines 34-35, consider revising the sentence structure to improve readability. A thorough proofreading for grammatical accuracy and fluency is recommended to ensure that the scientific content is communicated effectively.

Author Response

Comment 1; The title could be more concise. Consider simplifying it to enhance readability while retaining key information about the study's focus.

Response 1; We sincerely appreciate this suggestion. As recommended, we have revised the title to: "Breeding D1-Type Hybrid Japonica Rice for Diverse Upland Rainfed Environments." This revision maintains the core focus of the study while enhancing clarity and conciseness.

Comment 2; The introduction would benefit from a clearer structure. Consider outlining the specific research gap addressed by this study more explicitly at the end of the introduction.

Response 2: We thank the reviewer for this helpful observation. In revision, we have: (1) clearly identified the research gap in Lines 90-91, and (2) restructured the concluding paragraphs to better present our study objectives.

Comment 3; The introduction states the historical context of D1-type rice breeding. It would be helpful to include a clearer transition to the specific objectives of this study at the end of this section.

Response 3; Thank you for this helpful suggestion. We have revised the concluding paragraphs of the introduction to provide a clearer transition to our study objectives (see Lines 61-79)

Comment 4; In the methodology, please specify the criteria used for selecting the study sites. This information is crucial for understanding the environmental contexts of the findings. The authors mention "high-impact SNP variants." It would enhance clarity to define what constitutes a high-impact SNP in this context and how they were identified.

Response 4; We have addressed both points: (1) added detailed site selection criteria (Lines 394-395), and (2) defined 'high-impact SNPs' as variants called by ANNOVAR under default parameters (Line 409-412).

Comment 5; Line 68-69: The genetic analysis section discusses SNP mapping but lacks detail on the software or algorithms used. Please provide this information to improve reproducibility.

Response 5; In response, we have revised the genetic analysis section (Line 395-396) to specify that all SNP mapping analyses, including genotype identification, variant comparison, and visualization, were conducted using R statistical platform (version 4.4.2).

Comment 6; Line 95-96: When discussing the ecological adaptation of DHY615, consider including quantitative data or statistics to reinforce claims regarding its adaptability across different regions.

Response 6; We thank the reviewer for this important suggestion. We have incorporated our analysis by including quantitative yield performance data from multiple regions to demonstrate DHY615's ecological adaptation

Comment 7; Line 112-113: The results on grain yield performance could benefit from a comparison with other hybrid varieties. This context would strengthen the argument for DHY615's superiority.

Response 7; As suggested, We have expanded our yield evaluation to include water-saving and drought-resistant rice varieties (WDR) as comparators, demonstrating DHY615's competitive advantages

Comment 8; Line 145-146: The discussion of the SNP variations' functional impacts could be expanded. Consider providing specific examples of how these impacts translate to agronomic practices or outcomes.

Response 8; We appreciate the reviewer’s insightful suggestion regarding the functional implications of SNP variations. While this study identifies valuable genetic markers for targeted breeding (Lines 145–146), we acknowledge that further research is needed to fully characterize their agronomic impacts.

Comment 9; The methodology section lacks specific details about the statistical analyses used. Provide more information on the statistical tests applied to support the findings.

Response 9; Thank you for your valuable feedback. As per your suggestion, we have now included more specific details regarding the statistical analyses used in our study. The relevant information can be found on lines 396-397 in the methodology section, where we outline the specific statistical tests applied to support our findings.

Comment 10; Clarify the rationale behind the sample sizes used in both field trials and genetic analyses. Justification is important for assessing the robustness of the results.

Response 10; Thank you for your comment. We have clarified the rationale behind the sample sizes used in both the field trials and genetic analyses. The justification is now provided in lines (line 380).

Comment 11; In the genetic analysis section, provide more details on the SNP calling process, including thresholds for heterozygosity and the criteria for high-quality SNPs.

Response 11: Thank you for your helpful suggestion. As requested, we have now included additional details on the SNP calling process, including the thresholds for heterozygosity and the criteria for high-quality SNPs. This information can be found in lines 375-376 of the revised manuscript.

Comment 12; When discussing high-impact SNPs, consider providing additional context regarding how these SNPs were linked to specific agronomic traits.

Response 12; Thank you for your comment. We have clarified that the high-impact SNPs identified in this study are linked to candidate genes for specific agronomic traits. Future work will focus on these candidate genes related to agronomic practices.

Comment 13; Discuss how the findings of this study contribute to ecological sustainability in rice production, particularly in the context of changing climate scenarios.

Response 13; Thank you for your suggestion. Upon reflection, we find that the findings of this study do not directly address ecological sustainability in rice production, especially in the context of changing climate scenarios. Therefore, we have removed this discussion from the manuscript.

Comment 14; Include a comparative analysis of ‘DHY615’ with other commercially available hybrid rice varieties. This will enhance the manuscript's relevance to breeders and agricultural stakeholders.

Response 14; Thank you for your suggestion. As requested, we have now included a comparative analysis of ‘DHY615’ with water-saving and drought-resistant rice (WDR) varieties to enhance the manuscript’s relevance to breeders and agricultural stakeholders.

Comment 15; Acknowledge any limitations of the study in the discussion section. This will provide a balanced view and guide future research directions.

Response 15; Thank you for your suggestion. As requested, we have acknowledged the limitations of the study in the discussion section to provide a balanced view and guide future research directions. This can be found in lines 342-343 of the revised manuscript.

Comment 16; The genealogy figure could be enhanced with more annotations to clarify the contributions of different genetic sources more explicitly.

Response 16; Thank you for your valuable comment. While adding more annotations directly to the genealogy figure proved challenging, we have enhanced the figure legend with additional annotations to better clarify the contributions of different genetic sources.

Comment 17; Expand the discussion on the predicted functional impacts of SNP variations. How might these variations practically affect breeding strategies?

Response 17; Thank you for your comment. We have already discussed the predicted functional impacts of SNP variations and their practical implications for breeding strategies in the discussion section, along with supporting references

Comment 18; Provide more detailed metrics regarding grain quality assessments. Discuss how these metrics align with or differ from industry standards.

Response 18; Thank you for your suggestion. As requested, we have provided more detailed metrics regarding grain quality assessments. These metrics align with industry standards, and the grain quality meets the II-III class standards.

Comment 19; The results on field performance should include a discussion on the agronomic practices utilized during trials to better contextualize the findings.

Response 19; Thank you for your comment. We have added a discussion on the agronomic practices used during the trials, including e.g., farming practices, fertilization, pest control to better contextualize the findings.

Comment 20; Address the observed variability in yield performance across different locations and conditions. What factors may contribute to this variability?

Response 20; We appreciate your feedback. We have addressed the observed variability in yield performance across different locations and conditions in the revised manuscript. Potential factors contributing to this variability include soil quality, climate conditions, farming practices, and altitude.

Comment 21; Line 184-185: The conclusion mentions the potential for sustainable agricultural practices. It would be beneficial to suggest specific practices or strategies that could be implemented based on the study’s findings.

Response 21; We appreciate your suggestion. As requested, we have expanded the conclusion to include specific sustainable agricultural practices or strategies that could be implemented based on the study's findings. These strategies are supported by our technical protocol, which has already been incorporated into the manuscript.

Comment 22; Line 210-211: The authors acknowledge funding sources but do not discuss any potential conflicts of interest. Including a statement on conflicts of interest would improve transparency.

Response 22; We appreciate your insightful comment. As suggested, we have added a statement regarding conflicts of interest to improve transparency. This has been included in the revised manuscript.

Comment 21: The abstract is well-structured, but it would benefit from a clearer statement on how this study differs from previous work on D1-type hybrid japonica rice.

Response 21: Thank you for your feedback. The abstract has been revised to explicitly emphasize the unique aspects of this study in contrast to previous research on Yunnan-type japonica hybrid rice.

Comment 22: The introduction provides historical context but should better connect past research to the present study’s objectives.

Response 22: Thank you for your valuable suggestion. We have revised the concluding paragraphs of the introduction to establish a clearer connection between past research and the objectives of our present study.

Comment 23: The discussion section should explicitly relate the study's findings to real-world agricultural practices and climate resilience.

Response 23: Thank you for your insightful comment. We have revised the discussion section to explicitly relate our findings to real-world agricultural practices, particularly focusing on their implications for climate resilience in rice cultivation.

Comment 24: Some key terms, such as "heterozygous SNPs" and "functional allele genes," should be briefly explained for a broader audience.

Response 24: Thank you for pointing this out. The term "heterozygous SNPs" has been clarified in the methodology section (lines: 396-397). Additionally, the concept of "functional allele genes" is now described as being associated with essential agronomic traits, to ensure better understanding for a broader audience.

Comment 25: The manuscript would benefit from a clearer methodological explanation of how SNP variants were identified and classified.

Response 25: Thank you for your valuable feedback. We have addressed both points: (1) We have added more detailed site selection criteria (Lines: 375-376), and (2) We have clarified the definition of 'high-impact SNPs' as variants identified by ANNOVAR under the default parameters (Line :409-411).

Comment 26: The conclusion could be expanded to include specific recommendations for future research.

Response 26: Thank you for your suggestion. We have expanded the conclusion to include specific recommendations for future research (Lines :430-432).

Comment 27: Line 21-22: The phrase "thousands of heterozygous SNPs" is vague. Can you provide a precise number or a more refined estimate?

Response 27; Thank you for your comment. We have revised the phrase "thousands of heterozygous SNPs" to "6432 heterozygous SNPs" to provide a more precise estimate (Lines 25).

Comment 28: Line 25-26: The mention of "high UV radiation" as a stress factor is interesting. Can you provide references or specific data supporting this?

Response 28: Thank you for your suggestion. We have provided references supporting the mention of "high UV radiation" as a stress factor in Yun Province (Line: 90).

Comment 29: Line 30-31: The sentence “underscoring its potential for sustainable rice production” should include specific evidence from the study to support this claim.

Response 29: Thank you for your valuable feedback. We have revised the sentence to include specific evidence from the study that supports the claim of the potential for sustainable rice production. The revised statement now highlights the key findings that demonstrate this potential.

Comment 30: Line 36-38: The introduction states that WA and HL are primary CMS sources, but how does D1 compare in terms of practical advantages or disadvantages?

Response 30: Thank you for your insightful comment. We have revised the introduction to address the comparison of CMS D1-type with CMS WA and HL. The advantages of CMS D1-type include its key role in developing hybrid japonica and indica-japonica rice combinations such as ‘Yuza’, ‘Dianza’, ‘Dianheyou’, ‘Chunyou’, and ‘Yongyou’ series in China. However, its disadvantages include a relatively lower yield compared to WA, and HL.

Comment 31: Line 43-44: Consider rewording "a male sterile (MS) plant from the natural hybridization progeny" to improve clarity and readability.

Response 31: Thank you for your suggestion. We have revised the phrase "a male sterile (MS) plant from the natural hybridization progeny" to "a male sterile (MS) plant from the outcrossing hybridization progeny" to improve clarity and readability.

Comment 32: Line 47-48: The description of the pollen abortion process could be made clearer—perhaps with an additional figure or illustration.

Response 32: Thank you for your valuable suggestion. To improve clarity, we have provided additional references regarding the pollen abortion process, specifically regarding "stainable abortion" and "sterile gametophytes." We will also consider adding an illustration or figure to further clarify this process in the manuscript.

Comment 33: Line 49-50: The mention of atp6-orf79 and Rf1 genes is crucial; a short explanation of their significance for non-specialists would be helpful.

Response 33: Thank you for your helpful comment. We have added a brief explanation to clarify the significance of the atp6-orf79and Rf1 genes for non-specialists. In hybrid rice breeding, the CMS/Rf system is commonly used, where the CMS line carries mitochondrial toxin genes (atp6-orf79), and the restorer line carries the Rf gene (a PPR-type detoxification protein) that can restore fertility.

Comment 34: Line 51: The phrase “marked the establishment of D1-type hybrid rice breeding technology” should be revised to clarify the significance of this development.

Response 34: Thank you for your suggestion. We have revised the phrase to clarify the significance of this development: "In 1969, the naturally cross-pollinated male sterile line and its maintainer line were discovered. By 1973, the first Dian-type restorer line (Rf line) was successfully developed, marking the complete establishment of the Dian-type Three-Line Hybrid Rice System."

Comment 35: Line 325-326: Clarify the term “D1-type three-line japonica hybrid rice combination.” Does "D1-type" refer to a specific classification system? Adding a reference would be helpful.

Response 35: Thank you for your comment. We have clarified the term "D1-type three-line japonica hybrid rice combination" and added a reference to provide more context on the "D1-type" classification system.

Comment 36: Line 327-328: The sentence "In 2019, this variety was bestowed with the Gold Award..." lacks citation. Consider providing a reference or further explanation regarding the awarding institution. 

Response 36: Thank you for pointing this out. We have added a citation to provide more context regarding the Gold Award. Dianheyou 615 was awarded the Gold Medal based on sensory and palatability evaluations conducted by a panel of 31 rice experts, including Academician Wan Jianmin of the Chinese Academy of Engineering .(http://www.moa.gov.cn/xw/qg/202001/t20200108_6334536.htm).

Comment 37: Line 335-336: The explanation of ‘Nan615’ development could benefit from a more detailed description of its breeding process, including key genetic traits introduced from ‘Ansanbyeo’.

Response 37: Thank you for your suggestion. We have included a more detailed description of the breeding process for ‘Nan615’, highlighting key genetic traits introduced from ‘Ansanbyeo’ (a variety from Korea). These traits include lodging resistance and high yield, making it adaptable to direct seeding (Line:365).

Comment 38: Line 343-344: The phrase "several diversified regions with different elevations" is vague. It would be beneficial to specify the range of elevations tested to improve clarity.

Response 38: Thank you for your comment. We have revised the phrase "several diversified regions with different elevations" to "Yunnan Japonica Rice Growing Region (800-1600 masl)" to provide more clarity and specificity.

Comment 39: Line 347-348: Provide additional details or references for the field trial standards (DB42/T 1404-2018 and NY/T 2863-2015). If these are widely recognized standards, a brief description of their scope would be useful.

Response 39: Thank you for your suggestion. We will provide the references for the field trial standards (DB42/T 1404-2018 and NY/T 2863-2015) and include a brief description of their scope to clarify their relevance and recognition.

Comment 40: Line 349-350: The term “core on-farm demonstration locations” should be defined more precisely. What criteria were used to select these locations? Are they representative of typical rice-growing areas?

Response 40: Thank you for your comment. The term "core on-farm demonstration locations" refers to areas that are well-adapted to the local environment, ensuring stable yields across extensive rice cultivation areas. These locations were selected based on their representativeness of typical rice-growing conditions.

Comment 41: Line 420-421: The funding section mentions grants, but it would be helpful to briefly specify how these funds contributed to the study (e.g., data collection, experimental design, equipment).

Response 41: Thank you for your suggestion. We have revised the funding section to specify how the grants contributed to the study. Grant No. 202202AE09002102, Grant No. 202402AE090026, and Grant No. 202207AA110010 supported data collection and the investigation of traits, while the National Natural Science Foundation of China (Grant No. 31860108) contributed to the GSR40K SNP Check.

Comment 42: Line 426: The "Data Availability Statement" states that all data is contained within the article. However, the manuscript references supplementary tables (e.g., S1-S7). Consider specifying whether raw datasets are available elsewhere.

Response 42: Thank you for your comment. Currently, the raw SNP datasets have not been uploaded. However, if required, we are willing to upload them for further accessibility.

Comment 43: Line 417-419: In the “Author Contributions” section, clarify the specific roles of each author, especially regarding supervision and conceptualization. Were there any significant contributions from external collaborators?

Response 43: Thank you for your suggestion. We have revised the "Author Contributions" section to clarify the specific roles of each author, particularly regarding supervision and conceptualization. Additionally, we have addressed any significant contributions from external collaborators.

Comment 44: Line 429: The acknowledgment section lists names but does not mention specific contributions. Were these individuals involved in data collection, statistical analysis, or manuscript editing? Providing details would give better credit to their contributions.

Response 44: Thank you for your suggestion. We have revised the acknowledgment section to specify the contributions of each individual (Line:463-471). This provides clearer recognition of their roles in data collection, statistical analysis, and manuscript editing.

Reviewer 2 Report

Comments and Suggestions for Authors

In this manuscript, authors used the available data information, to analyze the phenotypic traits and genomic properties of a hybrid rice. Overall, authors did a good job, the manuscript was described clearly. Here, some minor comments list as below:

  1. It will be great, as if authors can provide the related data, such as field performance, of the parental strains of DHY615 and compare them.
  2. The data resource of haplotype map should be precisely cited.
  3. All figure legends are too abridged. They should be able to provide more details of explanation for that figure.
  4. Some of the font size in fig 2, 3, 4, and 5 are too small (and maybe also low resolution) to be clearly viewable. As if author can make a much better quality version that will be very good.

Author Response

Comment 1: It will be great, as if authors can provide the related data, such as field performance, of the parental strains of DHY615 and compare them.

Repones1: Thank you for your valuable comment. As requested, we have provided the field performance data of the parental strains and DHY615, along with a comparison of their field performance (lines 135-139). Additionally, yield-related characteristics per plant are included in Table S3.

Comment 2: The data resource of haplotype map should be precisely cited.

Respones2: Thank you for your suggestion. We have now included the precise citation for the data source of the haplotype map in the revised manuscript. This citation can be found in (Table 3).

Comment 3: All figure legends are too abridged. They should be able to provide more details of explanation for that figure.  

Response 3: Thank you for your valuable comment. We have provided additional explanations for Figures 1 to 6 in the revised manuscript for greater clarity.

Comment 4: Some of the font size in fig 2, 3, 4, and 5 are too small (and maybe also low resolution) to be clearly viewable. As if author can make a much better quality version that will be very good.

Response: As suggested, we have adjusted the font size in Figures 2 to 5 for better readability and improved the resolution to ensure clarity.
